# COVID-19 treatment combinations and associations with mortality in a large multi-site healthcare system

Dagan Coppock[1]*, Michael Baram[2], Anna Marie Chang[3], Patricia Henwood[3], Alan Kubey[4,5], Ross Summer[2], John Zurlo[1], Michael Li[6], Bryan Hess[1]

1 Division of Infectious Diseases, Department of Medicine, Thomas Jefferson University, Sidney Kimmel Medical College, Philadelphia, PA, United States of America, 2 Division of Pulmonary, Department of Medicine, Allergy, and Critical Care Medicine, Thomas Jefferson University, Sidney Kimmel Medical College, Philadelphia, PA, United States of America, 3 Department of Emergency Medicine, Thomas Jefferson University, Sidney Kimmel Medical College, Philadelphia, PA, United States of America, 4 Division of Hospital Medicine, Department of Medicine, Thomas Jefferson University, Sidney Kimmel Medical College, Philadelphia, PA, United States of America, 5 Division of Hospital Internal Medicine, Department of Internal Medicine, Mayo Clinic, Rochester, MN, United States of America, 6 Institute of Emerging Health Professions, Center for Digital Health and Data Science, Thomas Jefferson University, Philadelphia, PA, United States of America

* dagan.coppock@jefferson.edu

**Data Availability Statement:** A de-identified dataset is available in the Harvard Dataverse repository (DOI: 10.7910/DVN/DO66ZR).

## Abstract

### Introduction

During the early months of the COVID-19 pandemic, mortality associated with the disease declined in the United States. The standard of care for pharmacological interventions evolved during this period as new and repurposed treatments were used alone and in combination. Though these medications have been studied individually, data are limited regarding the relative impact of different medication combinations. The objectives of this study were to evaluate the association of COVID-19-related mortality and observed medication combinations and to determine whether changes in medication-related practice patterns and measured patient characteristics, alone, explain the decline in mortality seen early in the COVID-19 pandemic.

### Methods

A retrospective cohort study was conducted at a multi-hospital healthcare system exploring the association of mortality and combinations of remdesivir, corticosteroids, anticoagulants, tocilizumab, and hydroxychloroquine. Multivariable logistic regression was used to identify predictors of mortality for both the overall population and the population stratified by intensive care and non-intensive care unit admissions. A separate model was created to control for the change in unmeasured variables over time.

### Results

For all patients, four treatment combinations were associated with lower mortality: Anticoagulation Only (OR 0.24, p < 0.0001), Anticoagulation and Remdesivir (OR 0.25, p = 0.0031),

**Funding:** The authors received no specific funding for this work.

**Competing interests:** The authors have declared that no competing interests exist.

Anticoagulation and Corticosteroids (OR 0.53, p = 0.0263), and Anticoagulation, Corticosteroids and Remdesivir (OR 0.42, p = 0.026). For non-intensive care unit patients, the same combinations were significantly associated with lower mortality. For patients admitted to the intensive care unit, Anticoagulation Only was the sole treatment category associated with decreased mortality. When adjusted for demographics, clinical characteristics, and all treatment combinations there was an absolute decrease in the mortality rate by 2.5% between early and late periods of the study. However, when including an additional control for changes in unmeasured variables overtime, the absolute mortality rate decreased by 5.4%.

## Conclusions

This study found that anticoagulation was the most significant treatment for the reduction of COVID-related mortality. Anticoagulation Only was the sole treatment category associated with a significant decrease in mortality for both intensive care and non-intensive care patients. Treatment combinations that additionally included corticosteroids and/or remdesivir were also associated with decreased mortality, though only in the non-intensive care stratum. Further, we found that factors other than measured changes in demographics, clinical characteristics or pharmacological interventions accounted for an additional decrease in the COVID-19-related mortality rate over time.

## Introduction

As of July 2020, mortality related to COVID-19 was on the decline in the United States [1]. This may have been due to a change in treatment strategies, a change in patient characteristics, or a combination of both [2–4]. At that time, new or repurposed medications, such as remdesivir, corticosteroids, hydroxychloroquine, and tocilizumab were used, either alone or in combination, in an evolving standard of care for the treatment of COVID-19 [5–9].

As the pandemic has progressed, remdesivir, tocilizumab, and corticosteroids continue to receive attention for their potential benefit to mortality [5,10,11]. However, trial data from the World Health Organization suggest little benefit from the use of remdesivir in COVID-19 [12]. Similarly, there are clinical trials which demonstrate no mortality benefit of tocilizumab [13–15]. Furthermore, though clinical trials suggest a mortality benefit from dexamethasone in severely-ill patients [5], trials for other corticosteroids, including hydrocortisone and methylprednisolone, have not suggested a similar advantage [16–18].

Beyond these medications, the role of anticoagulation in COVID-19 has been of significant clinical interest. COVID-19 may predispose individuals to thrombotic disease due to a number of mechanisms [19]. Consequently, whether administered for prophylaxis or treatment, anticoagulation has been associated with a decrease in mortality in patients with COVID-19 [20,21].

Like other institutions, our healthcare system observed a decline in COVID-19-related mortality during the early months of the pandemic. Because of changing practice patterns in the treatment of COVID-19, we sought to explore the impact of different medication combinations on mortality. Using data obtained from our large, multi-hospital system, we examined the association between COVID-19-related mortality and various individual and combined treatments, including anticoagulants, remdesivir, corticosteroids, hydroxychloroquine and tocilizumab.

## Methods

### Ethics

This study was approved by the Jefferson University Institutional Review Board (IRB) (protocol approval #20E.846). It received a waiver for consent by the Jefferson University IRB.

### Setting and study population

The study was conducted as a retrospective, observational cohort study at Jefferson Health, a healthcare system that includes eleven inpatient facilities in southeastern Pennsylvania and southern New Jersey. The study cohort was comprised of COVID-19-positive patients admitted between March 1, 2020 and July 31, 2020. COVID-19 positivity was defined by the presence of at least one of the following: (1) a positive COVID-19 laboratory result during or within twenty-one days of the hospital admission; (2) COVID-19 infection added to a patient's electronic problem list during or within twenty-one days of the hospital admission; (3) an ICD-10 U07.1 code listed as one of the patient's final diagnoses from the admission.

### Outcomes

Mortality during admission was evaluated as the primary outcome. We explored this in two contexts. First, we examined the overall association of mortality in the setting of our principal exposure—medication treatment patterns. To accomplish this, we created a model that controlled for all available patient variables, including demographics, medications, admission characteristics, level of care, respiratory support, and comorbidities.

Second, we explored the change in mortality rates over time. For this, we used the model, as described above, to calculate predicted mortality rates for "early" and "late" periods of the pandemic. Early period admissions were defined as all admissions occurring between March 1, 2020 and April 30, 2020, while late period admissions were defined as all admissions occurring between May 1, 2020 and July 31, 2020. The demarcation of these periods was based on the May 1, 2020 issuance of the United States Food and Drug Administration's emergency use authorization for remdesivir [22].

### Exposure definitions

For the principal exposures of interest, several medications were evaluated for their association with mortality. These medications included remdesivir, corticosteroids, tocilizumab, anticoagulants, and hydroxychloroquine. Anticoagulant and corticosteroid use were defined as the use of a medication within the respective drug class during the admission regardless of the indication. In the case of anticoagulants, this included any use of anticoagulation, regardless of whether a medication was used for prophylaxis or systemic treatment.

To describe the treatment regimen administered to a given patient during admission, a "treatment combination" variable was created based upon the usage patterns of five medications—anticoagulants, corticosteroids, hydroxychloroquine, remdesivir and tocilizumab. To avoid multicollinearity, each patient was assigned to just one treatment combination category. All treatment combination categories with a case number less than fifty were grouped into a single category, labeled "all other combinations."

Other covariates of interest included patient demographics, specifically, age, gender, and race; admission characteristics, including length of stay and level of care; respiratory support / oxygenation, including mechanical ventilation, continuous positive airway pressure-bilevel positive airway pressure use, high flow nasal cannula (HFNC) use, and extracorporeal membrane oxygenation; and, comorbidities, including coronary artery disease / myocardial

infarction, congestive heart failure, peripheral vascular disease, cerebrovascular disease, dementia, chronic pulmonary disease, mild liver disease, moderate or severe liver disease, diabetes with complications, diabetes without complications, renal disease, cancer, metastatic carcinoma, and human immunodeficiency virus infection.

To estimate the impact of unmeasured variables over time, a binary "time period" variable was created. The variable included two categories—admission during the early period and admission during the late period, with periods defined as above.

### Statistical analysis

Descriptive statistics (means and proportions) were used to evaluate patient characteristics for the study period. Bivariate analyses were performed to test the association of mortality with variables of interest using Chi-square tests for categorical variables and t-tests for continuous variables.

Multivariable logistic regression was used to identify predictors of mortality for the overall population. To evaluate mortality for patients by severity of illness, we ran logistic models for both non-intensive care unit (ICU) admissions and ICU admissions. Models were created using a backward selection process for covariates. All treatment combination categories were included in the models, regardless of statistical significance. "No treatment" was used as the reference value for the treatment combination variable. The association strength of all independent variables was expressed as odds ratio (OR) estimates, 95% confidence intervals (CIs) and p-values. To address false significances from multiple testing, we calculated adjusted p-values by using the step-down Bonferroni method and the step-up Benjamini and Hochberg method. Statistical significance was set at p <0.05. Goodness-of-fit was evaluated using the Hosmer–Lemeshow test. In addition to unadjusted mortality rates, adjusted mortality rates were calculated for both early and late periods for two versions of the final multivariable model—one version that included the time-period variable as a predictor and one version that did not. All statistical analyses were performed using SAS 9.4 (SAS Institute Inc., Cary, NC).

### Results

During the study period, 4,351 patients were hospitalized with COVID-19, of which 664 patients diedand 3,687 patients were discharged alive. The average inpatient mortality rate for the entire period was 15.3%. However, overall mortality rates declined significantly from 21.3% in March 2020 to 8.8% in July 2020 (Fig 1). Rates declined for both ICU- and non-ICU-admitted patients and for patients less than sixty-five years old and sixty-five years old or greater.

Table 1 provides a comparison between patients who survived and those who died. A bivariate analysis of mortality was performed against key demographic and clinical variables. The average age of patients who survived was 62.5, whereas the average age for patients who died was 74.3 years-old (p-value < 0.0001). For patients who survived, the mean inpatient length of stay was 8.2 days (standard deviation of 9.0 days). For those who died, the mean length of stay was 9.8 days (standard deviation 9.3 days). Survival was significantly associated with gender, younger age, and White and Black/African American race, and Hispanic/Latino ethnicity. Overall, the patients who died had higher rates of cardiac, pulmonary, hepatic, and vascular disease. Regarding treatment, the use of anticoagulation and the use of remdesivir were not significantly different between those who died and those who survived. However, corticosteroids, hydroxychloroquine, and tocilizumab use were significantly higher in the cohort of patients who died.

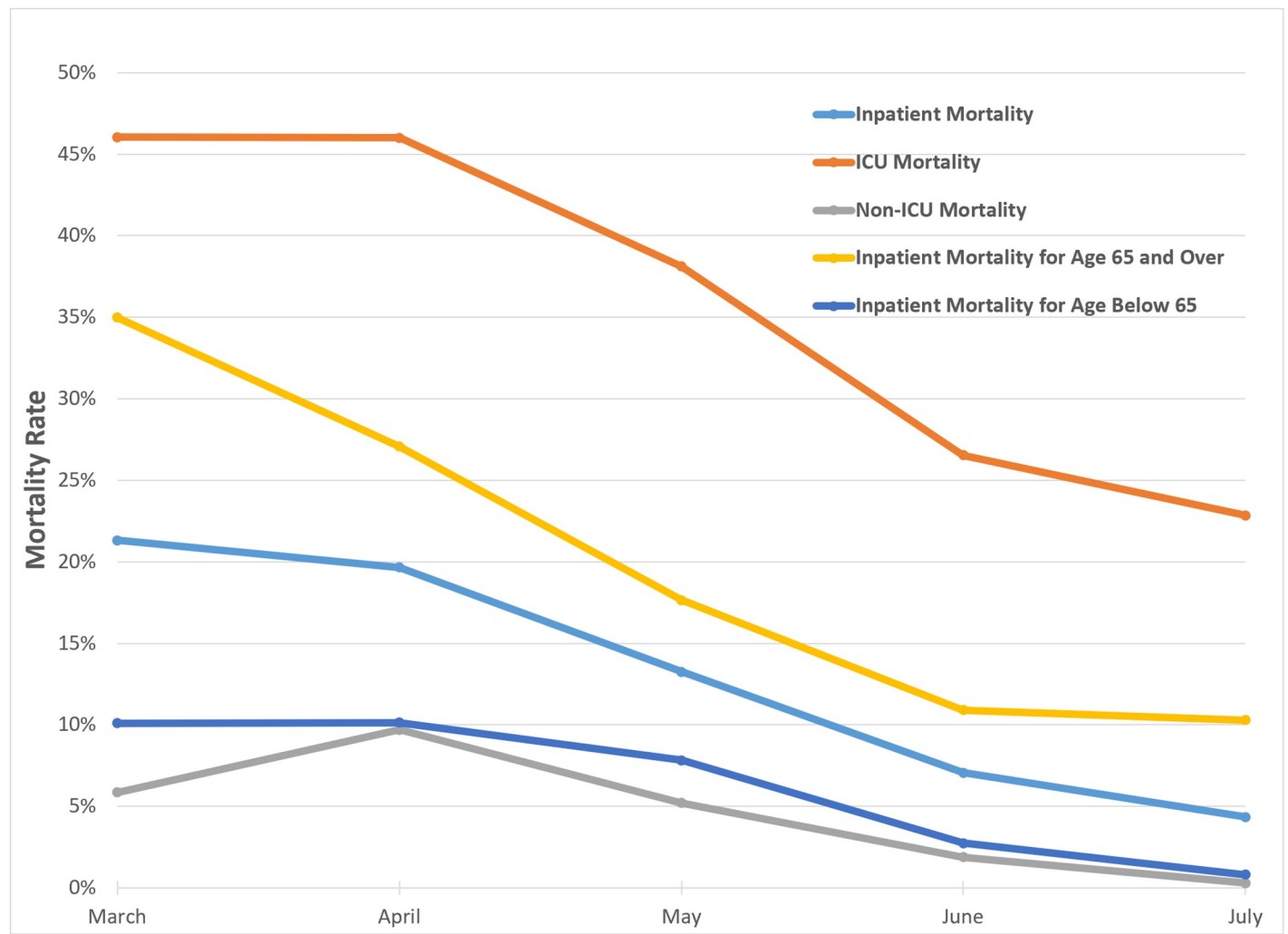

**Fig 1. Mortality Trends in COVID-19 admissions.** During the study period, overall mortality declined in the healthcare system. A decline was also seen regardless of stratification by age or level of care.

For the multivariable analysis, thirteen treatment categories were created based on observed treatment patterns as shown in Table 2. By case volume, the top three treatment combinations were "Anticoagulation Only" with 1,753 cases (40.3%), "Anticoagulation and Hydroxychloroquine" with 803 cases (18.5%) and "Anticoagulation and Corticosteroids" with 434 cases (10.0%). Ten treatment combinations contained less than fifty patients and were collapsed into a single category—"All Other Combinations." The "No Treatment" category had 258 patient cases (5.9%). The thirteen treatment categories were exclusive and contained a sufficient volume for analysis.

Major assumptions of multivariable logistic regression were met, given the binary dependent variables and the lack of multicollinearity among all independent variables. Variance inflation factors were all below 2. The C-statistic, which is same as the area under the receiver operating characteristic curve and measures the model's discrimination ability, was high at 0.921. Hosmer and Lemeshow tests demonstrated that the inpatient mortality model had a Chi-square of 11.4 and a high p-value of 0.183, indicating that the model was well calibrated and fit the data well. The study population was evaluated, both overall and stratified by level of

**Table 1. Demographic and clinical characteristics of COVID-positive patients by mortality status.**

| Variable | Survived (N = 3687) | | Died (N = 664) | | P-Value |
|---|---|---|---|---|---|
| | No. | % or Mean | No. | % or Mean | |
| **Age (years)** | | | | | |
| Less than 30 | 170 | 4.6% | 0 | 0.0% | < .0001 |
| 30–39 | 295 | 8.0% | 5 | 0.8% | < .0001 |
| 40–49 | 398 | 10.8% | 16 | 2.4% | < .0001 |
| 50–59 | 667 | 18.1% | 81 | 12.2% | 0.0002 |
| 60–69 | 752 | 20.4% | 132 | 19.9% | 0.7608 |
| 70–79 | 702 | 19.0% | 176 | 26.5% | < .0001 |
| 80–89 | 503 | 13.6% | 169 | 25.5% | < .0001 |
| 90 or greater | 200 | 5.4% | 85 | 12.8% | < .0001 |
| **Gender** | | | | | |
| Female | 1786 | 48.4% | 291 | 43.8% | 0.0284 |
| Male | 1901 | 51.6% | 373 | 56.2% | 0.0284 |
| **Race / Ethnicity** | | | | | |
| White | 1879 | 51.0% | 403 | 60.7% | < .0001 |
| Black / African American | 1295 | 35.1% | 189 | 28.5% | 0.0009 |
| Asian | 238 | 6.5% | 46 | 6.9% | 0.6499 |
| Hispanic / Latino | 166 | 4.5% | 6 | 0.9% | < .0001 |
| **Admission Characteristics** | | | | | |
| Inpatient Length of Stay (days) | 3687 | 8.2 | 664 | 9.8 | < .0001 |
| Intensive Care Unit Admission | 666 | 18.1% | 460 | 69.3% | < .0001 |
| **Respiratory Support / Oxygenation** | | | | | |
| Mechanical Ventilation | 310 | 8.4% | 371 | 55.9% | < .0001 |
| CPAP / BiPAP | 317 | 8.6% | 229 | 34.5% | < .0001 |
| High Flow Nasal Cannula | 747 | 20.3% | 284 | 42.8% | < .0001 |
| Extracorporeal Membrane Oxygenation | 11 | 0.3% | 9 | 1.4% | 0.0002 |
| **Treatment Received** | | | | | |
| Anticoagulation | 3399 | 92.2% | 609 | 91.7% | 0.6779 |
| Corticosteroid | 906 | 24.6% | 288 | 43.4% | < .0001 |
| Hydroxychloroquine | 1012 | 27.4% | 281 | 42.3% | < .0001 |
| Remdesivir | 320 | 8.7% | 59 | 8.9% | 0.8622 |
| Tocilizumab | 311 | 8.4% | 141 | 21.2% | < .0001 |
| Epoprostenol | 22 | 0.6% | 68 | 10.2% | < .0001 |
| Convalescent Plasma | 46 | 1.2% | 8 | 1.2% | 0.9296 |
| **Comorbidities** | | | | | |
| Coronary Artery Disease/Myocardial Infarction | 241 | 6.5% | 126 | 19.0% | < .0001 |
| Congestive Heart Failure | 559 | 15.2% | 199 | 30.0% | < .0001 |
| Peripheral Vascular Disease | 228 | 6.2% | 76 | 11.4% | < .0001 |
| Cerebrovascular Disease | 287 | 7.8% | 84 | 12.7% | < .0001 |
| Dementia | 621 | 16.8% | 188 | 28.3% | < .0001 |
| Chronic Pulmonary Disease | 814 | 22.1% | 183 | 27.6% | 0.0020 |
| Connective Tissue Disease | 92 | 2.5% | 28 | 4.2% | 0.0126 |
| Mild Liver Disease | 175 | 4.7% | 29 | 4.4% | 0.6707 |
| Moderate or Severe Liver Disease | 33 | 0.9% | 12 | 1.8% | 0.0325 |
| Diabetes without Complications | 1029 | 27.9% | 232 | 34.9% | 0.0002 |
| Diabetes with Complications | 514 | 13.9% | 151 | 22.7% | < .0001 |
| Renal Disease | 764 | 20.7% | 245 | 36.9% | < .0001 |

*(Continued)*

**Table 1.** (Continued)

| Variable | Survived (N = 3687) | | Died (N = 664) | | P-Value |
|---|---|---|---|---|---|
| | No. | % or Mean | No. | % or Mean | |
| Cancer | 149 | 4.0% | 47 | 7.1% | 0.0005 |
| Metastatic Carcinoma | 45 | 1.2% | 17 | 2.6% | 0.0073 |
| Moderate or Severe Liver Disease | 33 | 0.9% | 12 | 1.8% | 0.0325 |
| Human Immunodeficiency Virus Infection | 8 | 0.2% | 1 | 0.2% | 0.7289 |

CPAP/BiPAP, Continuous Positive Airway Pressure/Bilevel Positive Airway Pressure.

care, for predictors of mortality (Table 3). For the overall population, four medication categories had significant negative correlation estimates for mortality: Anticoagulation Only (OR 0.24, p < 0.0001), Anticoagulation and Remdesivir (OR 0.25, p = 0.0031), Anticoagulation and Corticosteroids (OR 0.53, p = 0.0263), and Anticoagulation and Corticosteroids and Remdesivir (OR 0.42, p = 0.026). Treatment combinations that included tocilizumab and/or hydroxychloroquine were not statistically significant.

Mortality was also evaluated based on level of care—non-ICU versus ICU patients (Table 3). The mortality rate was much higher in ICU patients compared to non-ICU patients —40.9% versus 6.3%, respectively. As in the model of all inpatients, the non-ICU model found that the same four treatment combinations were associated with lower mortality:

**Table 2. Medication combinations used to treat COVID-19.**

| Treatment Combination | Presence of the Treatment (Yes/No) | | | | | |
|---|---|---|---|---|---|---|
| | AC | CS | Remdesivir | HCQ | TCZ | Case Volume |
| No Treatment | No | No | No | No | No | 258 |
| AC Only | Yes | No | No | No | No | 1753 |
| AC and HCQ | Yes | No | No | Yes | No | 803 |
| AC and TCZ | Yes | No | No | No | Yes | 81 |
| AC and Remdesivir | Yes | No | Yes | No | No | 135 |
| AC and CS | Yes | Yes | No | No | No | 434 |
| AC and HCQ and TCZ | Yes | No | No | Yes | Yes | 76 |
| AC and CS and TCZ | Yes | Yes | No | No | Yes | 120 |
| AC and CS and HCQ | Yes | Yes | No | Yes | No | 255 |
| AC and CS and HCQ and TCZ | Yes | Yes | No | Yes | Yes | 112 |
| AC and CS and Remdesivir | Yes | Yes | Yes | No | No | 173 |
| AC and CS and Remdesivir and TCZ | Yes | Yes | Yes | No | Yes | 51 |
| All Other Combinations | No | No | No | Yes | No | 37 |
| | No | No | Yes | No | No | 2 |
| | No | Yes | No | No | No | 37 |
| | No | Yes | No | No | Yes | 1 |
| | No | Yes | No | Yes | No | 4 |
| | No | Yes | No | Yes | Yes | 1 |
| | No | Yes | Yes | No | No | 3 |
| | Yes | No | Yes | No | Yes | 10 |
| | Yes | No | Yes | Yes | No | 2 |
| | Yes | Yes | Yes | Yes | No | 3 |

AC, anticoagulation; CS, corticosteroids; HCQ, hydroxychloroquine; TCZ, tocilizumab.

**Table 3. Multivariable analysis of factors predicting mortality in COVID-19 patients.**

| Measure | All Inpatients (N = 4751) | | Non-ICU Inpatients (N = 3225) | | ICU Inpatients (N = 1526) | |
|---|---|---|---|---|---|---|
| | OR (95% CI) | P-value | OR (95% CI) | P-value | OR (95% CI) | P-value |
| **Race / Ethnicity** | | | | | | |
| Black / African American | 0.7 (0.5, 0.9) | 0.0078 | 0.6 (0.4, 1.0) | 0.0349 | — | — |
| Hispanic / Latino | 0.3 (0.1, 0.7) | 0.0095 | — | — | — | — |
| **Age (years)** | | | | | | |
| 40–49 | — | — | — | — | 3.8 (1.2, 12.7) | 0.0266 |
| 50–59 | 4.6 (2.5, 8.5) | < .0001 | — | — | 10.7 (3.7, 31.4) | < .0001 |
| 60–69 | 4.7 (2.6, 8.5) | < .0001 | 7.3 (2.6, 20.4) | 0.0001 | 8.5 (3.0, 24.3) | < .0001 |
| 70–79 | 10.2 (5.7, 18.5) | < .0001 | 12.3 (4.6, 33.1) | < .0001 | 19.7 (6.8, 56.9) | < .0001 |
| 80–89 | 21.7 (11.8, 40.1) | < .0001 | 29.2 (11.0, 77.8) | < .0001 | 29.1 (9.7, 86.8) | < .0001 |
| 90 or greater | 36.2 (18.7, 70) | < .0001 | 44.1 (16, 121.6) | < .0001 | 40.2 (11.6, 139.5) | < .0001 |
| **Admission Characteristics** | | | | | | |
| Inpatient Length of Stay | 0.91 (0.89, 0.92) | < .0001 | 0.93 (0.89, 0.96) | 0.0002 | 0.91 (0.90, 0.93) | < .0001 |
| ICU Admission | 2.7 (2.0, 3.8) | < .0001 | — | — | — | — |
| **Respiratory Support / Oxygenation** | | | | | | |
| Mechanical Ventilation | 19.6 (13.4, 28.8) | < .0001 | — | — | 13.4 (9.1, 19.9) | < .0001 |
| CPAP / BiPAP | 3.5 (2.6, 4.8) | < .0001 | 9.7 (5.8, 16.5) | < .0001 | 2.4 (1.7, 3.3) | < .0001 |
| High Flow Nasal Cannula | 1.7 (1.3, 2.2) | < .0001 | 3.3 (2.2, 4.8) | < .0001 | — | — |
| Extracorporeal Membrane Oxygenation | 18.8 (5.5, 64.1) | < .0001 | — | — | 9.1 (2.9, 28.5) | 0.0001 |
| **Comorbidities** | | | | | | |
| Myocardial Infarction | 1.9 (1.4, 2.7) | 0.0001 | 2.0 (1.2, 3.4) | 0.006 | 1.7 (1.1, 2.7) | 0.0124 |
| Peripheral Vascular Disease | 1.5 (1.1, 2.2) | 0.0242 | 1.8 (1.2, 2.6) | 0.0024 | — | — |
| Dementia | 1.6 (1.2, 2.1) | 0.002 | 2.3 (1.6, 3.4) | < .0001 | — | — |
| Diabetes Without Complications | 1.3 (1.0, 1.6) | 0.0427 | — | — | 1.4 (1.0, 1.9) | 0.0522 |
| Renal Disease | 2.0 (1.5, 2.5) | < .0001 | 1.7 (1.1, 2.4) | 0.0073 | 2.0 (1.5, 2.9) | < .0001 |
| Moderate or Severe Liver Disease | 3.4 (1.3, 9.2) | 0.0145 | 2.0 (1.1, 3.8) | 0.0303 | — | — |
| Metastatic Carcinoma | 3.6 (1.7, 7.7) | 0.0008 | — | — | 3.3 (0.9, 12.0) | 0.0643 |
| **Medication(s) Received** | | | | | | |
| AC only | 0.24 (0.15, 0.39) | < .0001 | 0.2 (0.11, 0.36) | < .0001 | 0.2 (0.1, 0.8) | 0.0231 |
| AC and HCQ | 0.7 (0.4, 1.1) | 0.1514 | 0.6 (0.3, 1.1) | 0.0978 | 0.5 (0.2, 1.9) | 0.3429 |
| AC and TCZ | 1.2 (0.6, 2.5) | 0.6832 | 1.5 (0.6, 4.0) | 0.3833 | 0.6 (0.1, 2.5) | 0.4354 |
| AC and Remdesivir | 0.25 (0.1, 0.63) | 0.0031 | 0.07 (0.01, 0.39) | 0.002 | 0.4 (0.1, 1.8) | 0.2128 |
| AC and CS | 0.53 (0.31, 0.93) | 0.0263 | 0.36 (0.17, 0.76) | 0.0074 | 0.5 (0.1, 1.8) | 0.2838 |
| AC and HCQ and TCZ | 0.7 (0.3, 1.6) | 0.3847 | 0.8 (0.1, 5.0) | 0.8566 | 0.6 (0.1, 2.2) | 0.4049 |
| AC and CS and TCZ | 1.2 (0.6, 2.6) | 0.5788 | 1.1 (0.3, 3.7) | 0.8645 | 0.7 (0.2, 2.8) | 0.6252 |
| AC and CS and HCQ | 0.8 (0.4, 1.5) | 0.4542 | 0.9 (0.3, 2.6) | 0.897 | 0.6 (0.2, 2.1) | 0.4345 |
| AC and CS and HCQ and TCZ | 0.8 (0.4, 1.6) | 0.4896 | 2.5 (0.5, 12.5) | 0.2596 | 0.5 (0.1, 2.0) | 0.3351 |
| AC and CS and Remdesivir | 0.42 (0.2, 0.9) | 0.026 | 0.07 (0.01, 0.59) | 0.014 | 0.6 (0.2, 2.4) | 0.4843 |
| AC and CS and Remdesivir and TCZ | 1.2 (0.5, 3.0) | 0.7222 | 0.3 (0.0, 3.3) | 0.3181 | 1.2 (0.3, 5.4) | 0.7668 |
| All Other Treatment Combinations | 1.3 (0.6, 2.8) | 0.4999 | 1.6 (0.6, 3.9) | 0.3103 | 1.0 (0.1, 8.0) | 0.9887 |

OR, odds ratio; CI, confidence interval; ICU, intensive care unit; AC, anticoagulation; CS, corticosteroids; HCQ, hydroxychloroquine; TCZ, tocilizumab.

Anticoagulation Only (OR 0.2, p < 0.0001); Anticoagulation and Remdesivir (OR 0.07, p = 0.002); Anticoagulation and Corticosteroids (OR 0.36, p = 0.0074); and, Anticoagulation and Corticosteroids and Remdesivir (OR 0.07, p = 0.014). Also, as with the unstratified model, all combinations containing tocilizumab and/or hydroxychloroquine were not statistically

**Table 4. Change in adjusted mortality between time periods.**

| Measure | Early Period: March 1, 2020 –April 30, 2020 | Late Period: May 1, 2020 –July 31, 2020 | Absolute Change In Mortality Rates Between Time Periods |
|---|---|---|---|
| Observed Mortality Rate | 19.9% | 10.1% | -9.8% |
| Mortality Rate, adjusted for all covariates except time-period | 7.4% | 4.9% | -2.5% |
| Mortality Rate, adjusted for all covariates including time-period | 7.4% | 2.0% | -5.4% |

significant. For ICU patients, Anticoagulation Only was the sole category associated with lower mortality (OR 0.2, p = 0.0231).

Table 4 provides a quantitative assessment of the impact of unexplained factors on the decline in mortality over time. Using the model-estimated parameters and patient data in two time periods, we calculated predicted mortality rates for early and late time periods. In row one of Table 4, observed mortality is reported for early and late time periods without risk adjustment for any of the covariates of interest. The observed mortality rate declined from 19.9% to 10.1%. In row two, mortality rates were predicted for the two time periods based on the multivariable model after adjusting for all covariates of interest, including demographics, clinical characteristics and treatment combinations. The risk-adjusted mortality rates for the two periods were 7.4% and 4.9%, respectively, demonstrating a 2.5% absolute reduction in the predicted mortality rate. When the time-period variable was added to the multivariable model, the absolute reduction in the predicted mortality rate between the two time periods was 5.4%.

## Discussion

Since the COVID-19 pandemic reached the United States, an accelerated response has been undertaken by healthcare providers and researchers [23]. In an ever-changing clinical environment, measuring outcomes, such as mortality, and their predictors can be a challenge, particularly given the rapid evolution of therapeutic strategies [24,25]. Through this study, we investigated mortality using data from a large healthcare system collected during the early months of the pandemic. The available data included not only patient demographics and clinical characteristics, but also details regarding medication treatment patterns.

Based upon our observations, anticoagulants, whether used for prophylaxis or treatment, were the most significant medications for reducing mortality in COVID-19. This is consistent with the current understanding of COVID-19 as a hypercoagulable state—one that is associated with thrombosis, which, in turn, can lead directly to mortality [26,27]. In this study, Anticoagulation Only was the sole treatment category associated with decreased mortality in both ICU- and non-ICU-admitted patients. Further, while combinations that included remdesivir and/or corticosteroids were associated with lower mortality in the overall population, the point estimates for the odds of mortality were worse compared to anticoagulation alone. When tocilizumab and/or hydroxychloroquine were included with regimens, regardless of the combination, there was no statistical benefit to mortality.

Like other institutions, we observed a shift in treatment patterns and patient demographics during the COVID-19 pandemic. In this study, we adjusted our mortality estimates not only for treatment patterns and patient characteristics, but also for time-period. When risk-adjusted for all available demographic and clinical factors, our model predicted an absolute decline in the mortality rate of 2.5%. However, we were concerned that there were other factors that may be contributing to the change in mortality. To address this concern, we included a time-period variable in the risk-adjusted model. When the time period variable was added to the model,

the absolute predicted mortality rate decreased by an additional 2.9%. The time-period variable was statistically significant, suggesting that there are other factors, unmeasured by our study, that have changed throughout the pandemic and have impacted mortality.

These unmeasured factors affecting mortality may be patient-centered, provider-centered or related to changes in the virus over time. Regarding patient factors, the population that was admitted early in the pandemic may be different than the population admitted late in the pandemic. It is unclear which differences are significant, though unmeasured aspects of health, particularly those related to socioeconomic status or functional capacity, may be contributing, all or in part, to the decline in mortality. Regarding provider factors, there have been a number of changes in standard of care, which were not included as variables in our study. For example, management protocols that favor HFNC and non-invasive ventilation over early intubation may have contributed to improved mortality [28,29]. Proning of patients, both on and off the ventilator, may also be a factor [30,31]. While our study strictly evaluated the presence or absence of treatments, there are other components of treatment management that have changed over the course of the pandemic. For instance, improved algorithms for anticoagulation [32] may have contributed to the mortality decline. Further, an increased willingness of providers to enter patients' rooms may have resulted in more attention to details, earlier detection of clinical decline, and resulting changes in treatment.

There are a number of limitations to our study. Though we attempted an indirect capture of unmeasured confounders, this is an observational study that did not include a number of important patient variables. Notably, our dataset did not include detailed demographic data such as occupation, household income, residence in a congregate setting such as a nursing home, or socioeconomic status. Further, it did not include clinical variables such as the patients' duration of symptoms at the time of presentation, their degree of hypoxia, or their laboratory values, such as C-reactive protein or D-dimer, all of which may predict mortality in COVID-related admissions [33–36].

Furthermore, our analysis of treatment combinations was broad. It only contained the binary value of a treatment's administration at least once during the admission. In our dataset, there was no information regarding treatment course, dose or timing. There were particular limitations in regards to our analysis of anticoagulation and corticosteroid administration. Anticoagulation, as defined by our study, encompassed a wide range of medications and dosages. Furthermore, anticoagulation was prescribed according to a combination of institutional guidelines and prescriber discretion, which likely contributed to dosing variability. National dosing strategies for anticoagulants are still being refined based on available trial data [37]. Likewise, corticosteroid use, as defined by our study, also contained different corticosteroids of variable potencies. Corticosteroid doses used for specific patients may not have been equivalent to the dexamethasone dosing that has become standard of care for COVID-19. Future studies will need to explore individual treatments in more detail.

In summary, we have presented a retrospective analysis of COVID-19-related mortality in patients hospitalized across a large healthcare system in the northeastern United States. Early in the pandemic, the mortality in the United States, broadly, and across Jefferson Health, specifically, underwent a decline. To understand this decline, we created models to test for predictors of mortality in hospitalized patients. This study finds that anticoagulation is an important contributor to the survival of patients hospitalized with COVID-19. Furthermore, this study suggests that the addition of remdesivir and corticosteroids, either alone or in combination, to anticoagulation may increase the odds of mortality in patients admitted to the ICU. Furthermore, the addition of tocilizumab to anticoagulation appears to nullify anticoagulation's statistically significant benefit to mortality in our cohort. This supports clinical trials arguing against the efficacy of this treatment. While our healthcare system's decline in mortality may

be, in part, related to the demographic and clinical factors explored in this study, there are likely additional factors that have contributed to the reduction in mortality. These will need to be explored through future analyses of our patient population.

## Author Contributions

**Conceptualization:** Dagan Coppock, Patricia Henwood, John Zurlo, Michael Li, Bryan Hess.

**Formal analysis:** Michael Li.

**Investigation:** Dagan Coppock, Patricia Henwood, John Zurlo, Bryan Hess.

**Methodology:** Dagan Coppock, Patricia Henwood, John Zurlo, Michael Li, Bryan Hess.

**Writing – original draft:** Dagan Coppock.

**Writing – review & editing:** Dagan Coppock, Michael Baram, Anna Marie Chang, Patricia Henwood, Alan Kubey, Ross Summer, John Zurlo, Michael Li, Bryan Hess.

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
