## [Decision Letter · Decision Letter 0]

24 Feb 2021

PONE-D-20-35523

COVID-19 Treatment Combinations and Associations with Mortality in a Large Multi-Site Healthcare System

PLOS ONE

Dear Dr. Coppock,

Thank you for submitting your manuscript to PLOS ONE. After careful consideration, we feel that it has merit but does not fully meet PLOS ONE’s publication criteria as it currently stands. Therefore, we invite you to submit a revised version of the manuscript that addresses the points raised during the review process.

We look forward to receiving your revised manuscript.

Kind regards,

Juan Carlos Lopez-Delgado, MD, PhD

Academic Editor

PLOS ONE

Journal Requirements:

2. In the ethics statement in the manuscript and in the online submission form, please provide additional information about the patient records used in your retrospective study. Specifically, please ensure that you have discussed whether all data were fully anonymized before you accessed them.

3. In statistical methods, please refer to any post-hoc corrections to correct for multiple comparisons during your statistical analyses. If these were not performed please justify the reasons. Please refer to our statistical reporting guidelines for assistance (https://journals.plos.org/plosone/s/submission-guidelines.#loc-statistical-reporting).

4. In your statistical analyses, please state whether you accounted for clustering by hospital. For example, did you consider using multilevel models?

Reviewers' comments:

Reviewer's Responses to Questions

**Comments to the Author**

1. Is the manuscript technically sound, and do the data support the conclusions?

Reviewer #1: Yes

Reviewer #2: Yes

2. Has the statistical analysis been performed appropriately and rigorously? 

Reviewer #1: I Don't Know

Reviewer #2: Yes

3. Have the authors made all data underlying the findings in their manuscript fully available?

Reviewer #1: Yes

Reviewer #2: Yes

4. Is the manuscript presented in an intelligible fashion and written in standard English?

Reviewer #1: Yes

Reviewer #2: No

5. Review Comments to the Author

Reviewer #1: This is a well constructed paper that reports the findings from a sound and interesting study in the era of COVID-19. The paper reassesses the impact of pharmacological interventions on COVID-19 mortality over time using a logistic regression model. Some specific areas to attend to are noted below:

1. One concern I have is whether assumptions of logistic regression were met. Please clarify how you validated the assumptions of logistic regression. It’s important that the data used meet major assumptions of logistic regression for the analysis to be valid.

2. In the abstract, the author says that ‘multivariate logistic regression’ was used, but ‘multivariate’ and ‘multivariable’ are two distinct approaches. In the results section the author says that “for all patients, four treatment combinations were associated with lower mortality: …, and anticoagulation, corticosteroids and remdesivir (OR 0.42, p <0.026).” I believe the author meant to say (OR 0.42, p=0.026).

3. In line 35, the authors report that ‘In the United States, mortality related to COVID-19 has been on the decline. This may be due to a change in treatment strategies, a change in patient characteristics, or a combination of both.” A reference/evidence to support this statement might be helpful. Does the data used for the study support shift in treatment patterns as well as shift in patient demographics?

4. Table 1 reports the mean/percentage of patient demographics and clinical characteristics. Reporting a measure of variability (standard deviation or standard error of the mean) for inpatient length of stays (days) or showing a general distribution of the length of stay might be helpful. The author may also recode patient length of stay as a categorical variable.

5.Table 4 reports the differences in mortality between the two time periods. I thought a figure showing variability over time would serve this paper well, especially in showing outcome changes over time (showing confounding). Again, a measure of variability (confidence interval or standard deviation) might be helpful in understanding the results.

6. How was the model validated? Reporting a goodness of fit statistic (for example, the Hosmer Lemeshow goodness of fit test) might be helpful in evaluating how well the proposed model fits or predicts a set of observations.

Reviewer #2: Coppock et al. shared experience of COVID treatment and decreased COVID-19 mortality in a large health care system. The finding of RDV and steroid combination was associated with lower mortality coincide with the clinical trial result.

It is hard to separate each COVID-19 treatment effect retrospectively, as disease severity and situation can be varied. The author simplified using the binary model and hinted association of anticoagulation use in morality.

It is reassuring to find the decrease in mortality over time in the paper, but a few additions may make the paper more informative.

(1) Was disease severity (mild, moderate, severe, critical) divided among the group? I assumed most hospitalized patients had the severe or critical disease. However, in-hospital transmission cases or patients with other comorbid conditions might have been admitted for the mild or moderate disease. One must be cautious looking into the morality as critical COVID patients may receive certain drugs versus mild COVID patients may not receive other than a routine stand of care prophylactic anticoagulation but have better outcomes.

(2) Over 90% received anticoagulation in COVID 19 patients. Were there specific criteria for anticoagulation use?

(3) Regarding the cause of death in the ICU group without anticoagulation, were there higher thromboembolic events?

(4) Consider including recent Multi-platform trial data regarding anticoagulation in the discussion section.https://www.nih.gov/news-events/news-releases/full-dose-blood-thinners-decreased-need-life-support-improved-outcome-hospitalized-covid-19-patients

In the discussion, the author mentioned that improved medication alone might not explain the decrease in mortality over time, which illustrates COVID 19 care's complexity in a real-life setting.

6. PLOS authors have the option to publish the peer review history of their article (what does this mean?). If published, this will include your full peer review and any attached files.

Reviewer #1: No

Reviewer #2: **Yes: **Dong Heun Lee

---

## [Author Response · Author response to Decision Letter 0]

12 Apr 2021

April 12, 2021

To the Editors of PLOS ONE:

We are resubmitting the manuscript “COVID-19 Treatment Combinations and Associations with Mortality in a Large Multi-Site Healthcare System” (Manuscript Number PONE-D-20-35523) for further review. Reviewer comments have been incorporated into the revised manuscript. We have itemized the comments and our responses to those comments below. Further, we have placed a de-identified dataset in the Harvard Dataverse repository at: https://doi.org/10.7910/DVN/DO66ZR.

Reviewer #1:

Comment: 1. One concern I have is whether assumptions of logistic regression were met. Please clarify how you validated the assumptions of logistic regression. It’s important that the data used meet major assumptions of logistic regression for the analysis to be valid.

Response: Major assumptions of logistic regression were met, including the use of binary dependent variables, lack of multicollinearity (variance inflation factor below 2) and independence of errors (data came from different patients). This is now addressed in the Results section of the manuscript (see paragraph starting at line 170).

Comment: 2. In the abstract, the author says that ‘multivariate logistic regression’ was used, but ‘multivariate’ and ‘multivariable’ are two distinct approaches. In the results section the author says that “for all patients, four treatment combinations were associated with lower mortality: …, and anticoagulation, corticosteroids and remdesivir (OR 0.42, p <0.026).” I believe the author meant to say (OR 0.42, p=0.026).

Response: Thank you for this great point. Multivariable logistic regression is the correct terminology. We have revised the manuscript accordingly. Regarding the p-value, thank you for detecting our error. The results now read OR 0.42, p=0.026 instead of OR 0.42, p <0.026.

Comment: 3. In line 35, the authors report that ‘In the United States, mortality related to COVID-19 has been on the decline. This may be due to a change in treatment strategies, a change in patient characteristics, or a combination of both.” A reference/evidence to support this statement might be helpful. Does the data used for the study support shift in treatment patterns as well as shift in patient demographics?

Response: References are now provided to support the above statement. Furthermore, like other institutions, we have observed a shift in treatment patterns and patient demographics during the Covid-19 pandemic. A statement related to this is now included in the revised discussion section (see line 243). 

Comment: 4. Table 1 reports the mean/percentage of patient demographics and clinical characteristics. Reporting a measure of variability (standard deviation or standard error of the mean) for inpatient length of stays (days) or showing a general distribution of the length of stay might be helpful. The author may also recode patient length of stay as a categorical variable.

Response: We have now provided the standard deviation for length of stay stratified by outcome in the revised Results section.

Comment: 5. Table 4 reports the differences in mortality between the two time periods. I thought a figure showing variability over time would serve this paper well, especially in showing outcome changes over time (showing confounding). Again, a measure of variability (confidence interval or standard deviation) might be helpful in understanding the results.

Response: Table 4 has been revised. The observed mortality rate for the two time periods is now included (i.e., the non-risk adjusted mortality is now included). When comparing two time periods (versus repeated measurements), confidence intervals are not provided unless bootstrapping is performed. Since our study was a retrospective empirical study, we did not feel bootstrapping was necessary. 

Comment: 6. How was the model validated? Reporting a goodness of fit statistic (for example, the Hosmer Lemeshow goodness of fit test) might be helpful in evaluating how well the proposed model fits or predicts a set of observations.

Response: Because the purpose of our study was to identify correlated risk factors, not to predict mortality, we did not perform training and holdout validations. However, Hosmer and Lemeshow tests did show a good fit to our model and a statement related to these results is now included in the revised manuscript (see line 174).

Reviewer #2: 

Comment: (1) Was disease severity (mild, moderate, severe, critical) divided among the group? I assumed most hospitalized patients had the severe or critical disease. However, in-hospital transmission cases or patients with other comorbid conditions might have been admitted for the mild or moderate disease. One must be cautious looking into the mortality as critical COVID patients may receive certain drugs versus mild COVID patients may not receive other than a routine stand of care prophylactic anticoagulation but have better outcomes.

Response: This is an excellent point. To address this issue, we performed multivariable regression analyses on both ICU and non-ICU patient populations. We have edited our Methods section (see line 114) to clarify this approach.

Comment: (2) Over 90% received anticoagulation in COVID 19 patients. Were there specific criteria for anticoagulation use?

Response: The decision to use anticoagulation was a combination of institutional thromboprophylaxis guidelines and individual provider discretion. A statement related to this is now included in the Discussion section of the revised manuscript (see line 281). 

Comment: (3) Regarding the cause of death in the ICU group without anticoagulation, were there higher thromboembolic events?

Response: This study did not explore this endpoint. However, other investigators at our institution are currently examining the association between anticoagulation and thromboembolic events at Jefferson Health. 

Comment: (4) Consider including recent Multi-platform trial data regarding anticoagulation in the discussion section.https://www.nih.gov/news-events/news-releases/full-dose-blood-thinners-decreased-need-life-support-improved-outcome-hospitalized-covid-19-patients

Response: We have now included this in the revised discussion (see line 282). 

General Editor (from initial submission):

Comment: 1. Please ensure that your manuscript meets PLOS ONE's style requirements, including those for file naming. The PLOS ONE style templates can be found at

Response: The manuscript has been formatted to meet PLOS ONE’s style requirements.

Comment: 2. In the ethics statement in the manuscript and in the online submission form, please provide additional information about the patient records used in your retrospective study. Specifically, please ensure that you have discussed whether all data were fully anonymized before you accessed them.

Response: The ethics statement has been clarified to reflect that this project was evaluated by the Jefferson University Institutional Review Board and received a waiver of authorization for use of protected health information (control number 20E.846). 

Comment: 3. In statistical methods, please refer to any post-hoc corrections to correct for multiple comparisons during your statistical analyses. If these were not performed please justify the reasons. Please refer to our statistical reporting guidelines for assistance (https://journals.plos.org/plosone/s/submission-guidelines.#loc-statistical-reporting).

Response: To address false significances from multiple testing, we calculated adjusted p-values by using (1) the step-down Bonferroni method and (2) the linear step-up method of Benjamini and Hochberg that that controls the false discovery rates. This is now stated in our Methods section (see line 120). 

Comment: 4. In your statistical analyses, please state whether you accounted for clustering by hospital. For example, did you consider using multilevel models?

Response: Per common practice, we pooled patients across multiple facilities and did not perform multilevel modeling or fixed-random effect model. Identifying facility cluster effects was not one of our initial study objectives. However, it would be interesting to explore the hospital effects and understand how patient and treatment risks can be different across facilities. We consider this a good further research topic.

Comment: 5. We note that you have indicated that data from this study are available upon request. PLOS only allows data to be available upon request if there are legal or ethical restrictions on sharing data publicly. For information on unacceptable data access restrictions, please see http://journals.plos.org/plosone/s/data-availability#loc-unacceptable-data-access-restrictions.

Response: A deidentified data set has been placed in the Harvard Dataverse repository. It may be reviewed at https://doi.org/10.7910/DVN/DO66ZR. 

General Editor (from re-submission):

Comment: 1) Thank you for including your ethics statement on the online submission form: 

"This study was approved by the Jefferson University Institutional Review Board (protocol approval #20E.846). The study received a waiver for consent by the Jefferson University Institutional Review Board." To help ensure that the wording of your manuscript is suitable for publication, would you please also add this statement at the beginning of the Methods section of your manuscript file.

Response: We have moved the ethics statement to the beginning of the Methods section. 

Comment: Thank you for your e-mail to PLOS ONE following our queries regarding data availability… In light of this we are unable to sign the DUA presented. Furthermore, we would be grateful if you could provide some clarification on the following:

2.1) Whether you are still able to make the data available in a public repository (preferred) or as part of the submitted article.

2.2) If there are restrictions to sharing the data as above, please indicate who is imposing these restrictions…

2.3) …and why such restrictions are imposed on the data.

Response: We have added the de-identified dataset to the Harvard Dataverse repository. There are no restrictions on sharing the de-identified data. 

Thank you for your consideration of this revised manuscript. We hope these revisions address any concerns. Please contact me with any further questions. 

Sincerely,

Dagan Coppock, MD, MSCE

---

## [Decision Letter · Decision Letter 1]

19 May 2021

COVID-19 Treatment Combinations and Associations with Mortality in a Large Multi-Site Healthcare System

PONE-D-20-35523R1

Dear Dr. Coppock,

We’re pleased to inform you that your manuscript has been judged scientifically suitable for publication and will be formally accepted for publication once it meets all outstanding technical requirements.

Kind regards,

Juan Carlos Lopez-Delgado, MD, PhD

Academic Editor

PLOS ONE

Additional Editor Comments (optional):

Dear authors,

All questions from the previous reviewers have been addressed. All apologies for the delay, it was very difficult to find reviewers for the present manuscript in pandemic times.

However, remember PLOS One do not have editorial services and your manuscript has to be in perfect scientific english. Please, review in order to avoid any editorial mistake or minor error.

Take care & keep safe in these pandemic times. Warm regards,

Juan Carlos

Reviewers' comments:

Reviewer's Responses to Questions

**Comments to the Author**

1. If the authors have adequately addressed your comments raised in a previous round of review and you feel that this manuscript is now acceptable for publication, you may indicate that here to bypass the “Comments to the Author” section, enter your conflict of interest statement in the “Confidential to Editor” section, and submit your "Accept" recommendation.

Reviewer #2: All comments have been addressed

Reviewer #3: All comments have been addressed

2. Is the manuscript technically sound, and do the data support the conclusions?

Reviewer #2: Yes

Reviewer #3: Yes

3. Has the statistical analysis been performed appropriately and rigorously? 

Reviewer #2: Yes

Reviewer #3: Yes

4. Have the authors made all data underlying the findings in their manuscript fully available?

Reviewer #2: Yes

Reviewer #3: Yes

5. Is the manuscript presented in an intelligible fashion and written in standard English?

Reviewer #2: Yes

Reviewer #3: Yes

6. Review Comments to the Author

Reviewer #2: Author appropriately addressed the comments and revised manuscript. I would like to congratulate the authors for sharing important findings in timely manner.

Reviewer #3: The revision of the present manuscript has been delayed due to the difficulties in finding appropriate reviewes in these pandemic times. All apologies for this delay. All concerns have been addressed. Remember to review english scientific language before editing.

7. PLOS authors have the option to publish the peer review history of their article (what does this mean?). If published, this will include your full peer review and any attached files.

Reviewer #2: **Yes: **Dong Heun Lee

Reviewer #3: No

---

## [Editor Report · Acceptance letter]

25 May 2021

PONE-D-20-35523R1 

COVID-19 Treatment Combinations and Associations with Mortality in a Large Multi-Site Healthcare System 

Dear Dr. Coppock:

I'm pleased to inform you that your manuscript has been deemed suitable for publication in PLOS ONE. Congratulations! Your manuscript is now with our production department. 

Kind regards, 

on behalf of

Dr. Juan Carlos Lopez-Delgado 

Academic Editor

PLOS ONE